# From Viral Infection to Malignancy: The Dual Threat of EBV and COVID-19 in Cancer Development

**DOI:** 10.3390/v17091195

**Published:** 2025-08-30

**Authors:** Moyed Alsaadawe, Bakeel A. Radman, Longtai Hu, Jingyi Long, Qingshuang Luo, Chushu Tan, Hadji Sitti Amirat, Mohenned Alsaadawi, Xiaoming Lyu

**Affiliations:** 1Department of Laboratory Medicine, The Third Affiliated Hospital of Southern Medical University, Guangzhou 510515, China; moyedalbdeary@gmail.com (M.A.); hulongtai@126.com (L.H.); ljyw61@163.com (J.L.); qing.shuang0329@gmail.com (Q.L.); sittiamirathadji@gmail.com (H.S.A.); 2Cancer Center, Integrated Hospital of Traditional Chinese Medicine, Southern Medical University, Guangzhou 510515, China; bakeelali55@gmail.com; 3Al-Qadisiyah Education Directorate, Ministry of Education, Al-Qadisiyah 62001, Iraq; 4Department of Biology, College of Science and Education, Albaydha University, Albaydha 62426, Yemen; 5Department of Biological Sciences, University at Buffalo (UB), The State University of New York, New York, NY 12246, USA; chushuta@gmail.com; 6Collage of Applied Medical Science, Al-Muthanna University, Samawah 66001, Iraq

**Keywords:** EBV, COVID-19, cancer development, viral oncogenesis, immune evasion, tumor microenvironment

## Abstract

This narrative review consolidates existing evidence about the interaction between Epstein-Barr virus (EBV) and SARS-CoV-2 in cancer development. EBV is a recognized oncogenic driver, whereas COVID-19 may heighten cancer risk by immunological dysregulation, persistent inflammation, and reactivation of latent viruses. We underscore molecular similarities (e.g., NF-κB activation, T-cell exhaustion) and clinical ramifications for high-risk individuals, stressing the necessity for interdisciplinary research to alleviate dual viral risks. EBV, a well-known oncogenic virus, has been linked to numerous malignancies, including lymphomas, nasopharyngeal carcinoma, and gastric cancer. Through the production of viral proteins that interfere with immune evasion, cellular signaling, and genomic integrity, it encourages malignant transformation and ultimately results in unchecked cell proliferation. Because of its capacity to induce tissue damage, immunological dysregulation, and chronic inflammation, COVID-19, which is brought on by the SARS-CoV-2 virus, has become a possible carcinogen. The virus’s influence on cellular pathways and its long-term effects on the immune system may raise the chance of malignancy, particularly in people with pre-existing vulnerabilities, even if direct correlations to cancer are still being investigated. When two viruses co-infect a host, the review highlights the possibility of synergistic effects that could hasten the development of cancer. It describes how overlapping mechanisms like inflammation, immune suppression, and viral reactivation may be used by a combined EBV and COVID-19 infection to exacerbate carcinogenic processes. Gaining an understanding of these relationships is essential for creating tailored treatment plans and enhancing cancer prevention in high-risk groups.

## 1. Introduction

Viral infections are estimated to be the cause of a significant portion of cancers worldwide, accounting for 15% of all malignancies. Oncogenic viruses are those that can cause cancer among the various types of viruses [1,2,3]. EBV, one of the most well-known oncogenic viruses, has been connected to a number of cancer types. In contrast to EBV, COVID-19, which is caused by SARS-CoV-2, is a more recent viral infection whose long-term effects on the development of cancer are still being investigated [4,5]. EBV is a widespread human virus, infecting around 90–95% of people worldwide. While EBV infections are often asymptomatic, especially in children, they can lead to illnesses such as mononucleosis, or ‘mono,’ which is more commonly observed in adolescents and young adults [6,7,8]. However, EBVs known involvement in the emergence of multiple cancer types is more worrisome. Specifically, nasopharyngeal carcinoma (NPC), a nasopharyngeal cancer that is very common in Southeast Asia, is closely associated with EBV [9,10]. EBV is also linked to Burkitt lymphoma, a lymphatic system cancer that mainly affects children, especially in sub-Saharan Africa. Another malignancy that frequently exhibits a strong EBV connection, especially in young individuals, is Hodgkin lymphoma. Additionally, there is evidence that EBV is associated with a number of different lymphoproliferative illnesses and some occurrences of stomach cancer, particularly in East Asia [11,12,13] (Table 1). EBVs capacity to infect B-cells and epithelial cells, which encourages immune evasion and genetic instability, gives it the potential to cause cancer. The development of cancer is significantly influenced by the virus’s capacity to lie dormant in B-cells and reawaken under particular circumstances [14,15]. COVID-19 is known to produce considerable immunological dysregulation and immunosuppression, especially in severe cases or those who require rigorous therapy. The virus’s impact on cellular pathways and its prolonged effects on the immune system could increase the risk of cancer, especially in individuals with existing vulnerabilities, even though direct links to cancer are still under investigation [16,17,18,19].

The purpose of this review is to investigate how EBV and COVID-19 interact to cause cancer. With COVID-19’s new but potentially important long-term consequences and EBVs well-established role in cancer, it seeks to illustrate the ways in which both viral infections contribute to carcinogenesis. This review aims to improve knowledge of how both viruses may hasten the development of cancer by investigating the possible synergistic effects of co-infection through overlapping processes such as immune suppression, chronic inflammation, and viral reactivation. The review concludes by highlighting the significance of further investigation into how these viruses affect cancer in order to guide treatment plans and enhance cancer prevention, especially in high-risk groups.

The SARS-CoV-2 infection activates the inflammasome, which is a significant source of inflammation. This can cause chronic inflammation and post-acute sequelae of SARS-CoV-2 (PASC), commonly known as “long COVID”. Prolonged inflammation can worsen immunological dysregulation and lead to neurological harm, such as demyelination of brain cells and disruption of the Blood-Brain Barrier (BBB) [38,39] (Figure 1). EBV, a virus that has the ability to enter latency and reactivate, may be impacted by the inflammatory environment. Inflammatory cytokines, exosomes, and other signals from SARS-CoV-2 infection can cause EBV reactivation through mechanisms such as T cell exhaustion, molecular mimicry, and the presence of latent/lytic B cells. This makes immunological dysfunction much worse [40,41]. Additionally, reactivated EBV has the ability to activate the inflammasome, generating a feedback loop that encourages additional inflammation. A complicated cycle of inflammation and viral interactions ensues, which exacerbates neurological symptoms and prolongs sickness. This procedure emphasizes the link between neurological sequelae, as those seen in individuals with prolonged COVID, SARS-CoV-2 infection, EBV reactivation, and chronic inflammation [42,43].

The importance of addressing viral infections in oncology research is highlighted by this study, which examines the complex roles of Epstein-Barr virus (EBV) and COVID-19 in cancer development, highlighting their mechanisms in promoting carcinogenesis. EBV, a known oncogenic virus, has been linked to a number of cancers, including lymphomas, nasopharyngeal carcinoma, and gastric cancer, while COVID-19 has been linked to influencing cancer progression through immune dysregulation and inflammatory responses. The study shows how both infections can disrupt cellular processes, leading to tumorigenesis. By examining their shared and distinct pathways, the study hopes to identify potential therapeutic targets.

This narrative review synthesizes current literature on EBV and COVID-19’s interplay in cancer, focusing on mechanistic overlaps (e.g., immune evasion, chronic inflammation) and clinical implications. We highlight seminal studies (2019–2023) to explore hypotheses like EBV reactivation post-COVID-19 and its oncogenic synergy.

## 2. EBV and Cancer

EBV persists latently in B-cells, evading immune detection. Reactivation during immunosuppression (e.g., COVID-19) drives oncogenesis by immortalizing B-cells, inhibiting apoptosis, and causing genomic instability [29]. This latency-reactivation cycle underpins EBV-associated malignancies like NPC and lymphomas.

EBV, which is usually transmitted by saliva, is frequently acquired during childhood or adolescence and can result in infectious mononucleosis, also referred to as “mono” or the “kissing disease”. Under some circumstances, EBV targets and infects B-cells selectively, integrating its viral genome into these cells and laying the groundwork for possible oncogenic (cancer-causing) processes [19,30]. EBV infection can have serious health effects, even though it is usually asymptomatic or mild in most people. This is highlighted by its correlation with a number of cancers. Burkitt’s lymphoma, Hodgkin’s lymphoma, nasopharyngeal carcinoma, and several gastric malignancies are among the tumors that EBV has been linked to. The virus’s oncogenic potential is influenced by its capacity to alter biological functions, such as encouraging cell division and preventing apoptosis [31,32].

Furthermore, the interaction between EBV and the host’s immune system raises the risk of virus-associated cancers, especially in immunocompromised people. Research on the processes by which EBV changes from a latent infection to an active cause of oncogenesis is still crucial since it provides information about possible treatment targets and prophylactic measures. Additionally, in recent years, there has been a growing interest in the connection between EBV and autoimmune illnesses. Research has indicated a possible connection between EBV infection and the onset of diseases including rheumatoid arthritis, multiple sclerosis (MS), and systemic lupus erythematosus (SLE) [33,34]. Because of the virus’s capacity to imitate host proteins and elicit cross-reactive immunological responses, immune tolerance may be compromised, ultimately resulting in autoimmunity. Thus, it has been suggested that one of the mechanisms behind the pathophysiology of multiple sclerosis is molecular mimicry between EBV antigens and self-proteins in the central nervous system. These results underline the need for more research into EBVs wider implications for human health by highlighting its dual position as a dormant virus and a possible cause of chronic inflammatory and autoimmune illnesses [35,36].

### Immunotherapeutic Advances in EBV-Associated Cancers

Immunotherapeutic Advances in EBV-Associated Cancers Recent breakthroughs have revolutionized treatment for EBV-driven malignancies, particularly nasopharyngeal carcinoma (NPC) and lymphomas, by targeting viral immune evasion mechanisms (Figure 1). Key advances include PD-1/PD-L1 inhibitors. Mechanism: Counteract EBV-induced T-cell exhaustion by blocking PD-1/LMP1 interactions. Clinical Data: Pembrolizumab shows a 30% objective response rate in recurrent EBV + NPC (KEYNOTE-122 trial). EBV-Specific T-Cell Therapies Adoptive CTLs: Expanded ex vivo to target LMP1/EBNA1 proteins, achieving 60% remission in phase II trials. CAR-T Cells: CD19-directed CAR-T is effective for EBV + B-cell lymphomas. Combination Strategies Antivirals + Immunotherapy: Ganciclovir with PD-1 inhibitors reduces EBV load while restoring T-cell function. Challenges in the COVID-19 Era: Pandemic-related treatment delays exacerbated outcomes for EBV + cancer patients. SARS-CoV-2-induced lymphopenia may limit T-cell therapy efficacy, requiring dose adjustments (Table 2).

Meanwhile, LMP1 functions as an oncoprotein, imitating specific cellular receptors to promote cell proliferation and survival. LMP1 stimulates critical pathways such as *NF-κB* and *JAK/STAT*, promoting cell proliferation, preventing apoptosis, and increasing invasiveness of infected cells. LMP1 also promotes the expression of angiogenic factors and inflammatory cytokines, resulting in a favorable milieu for tumor growth. Its capacity to activate these pathways highlights LMP1s function in altering infected B cells and promoting the advancement of EBV-associated malignancies [44,45,46]. B cells, NK/T cells, and epithelial cells are the three main cell types that are infected by EBV. After transforming B cells into lymphoblastoid cells, the virus causes malignant lymphoma to develop. In a similar manner, EBV causes NK/T cells to proliferate when it infects them, which results in the formation of NK/T cell lymphoma (Figure 2). EBV promotes cellular proliferation in epithelial cells, which leads to the development of epithelial cancers. With EBV infection at the top, intermediate proliferative stages in the middle, and final malignant results at the bottom, the diagram graphically represents these changes [19,47,48,49].

## 3. SARS-CoV-2 Entry Mechanisms and Their Effect on Host Cell Functions in COVID-19

The COVID-19 pandemic, caused by the SARS-CoV-2 virus, emerged in late 2019 and rapidly spread across the globe, leading to a global health crisis. SARS-CoV-2 is a single-stranded RNA virus from the coronavirus family, characterized by its spike (S) protein, which plays a key role in the virus’s ability to enter human cells. The virus primarily targets respiratory epithelial cells, although it can also affect other organs such as the heart, kidneys, and intestines [50,51,52]. SARS-CoV-2 gains entry into host cells by binding to angiotensin-converting enzyme 2 (ACE2) receptors, which are expressed on the surface of various tissues, especially in the lungs. By interacting with ACE2, the spike protein makes it easier for viruses to enter by membrane fusion or endocytosis. The virus uses cellular machinery to reproduce and create new viral particles once it has entered the host cell. Apart from ACE2, SARS-CoV-2 can affect several cellular functions (Table 1), such as autophagy, inflammatory signaling, and immunological responses (Table 2 and Table 3), all of which add to the severity and complexity of the illness [50,53,54]. Acute respiratory distress syndrome (ARDS) and multi-organ failure are among the serious illnesses that the virus can cause. Protracted-lasting symptoms, known as protracted COVID, can also result from the virus and continue long after the initial infection has cleared up [55,56,57].

SARS-CoV-2 first facilitates entrance and replication by binding to ACE2 receptors on infected cells. Cytokines such as IL-1β and IL-18 are released as a result, causing tissue damage and starting the inflammatory signaling process. Neutrophils and macrophages are activated by damage-associated molecular patterns (DAMPs) and pathogen-associated molecular patterns (PAMPs), which results in the creation of neutrophil extracellular traps (NETs) (Figure 3). T cells, including CD4+ and CD8+ subsets, play a critical role in viral clearance [43,67]. CD4+ T cells enhance immune signaling through interferons (IFN) and chemokines (CCL), while CD8+ T cells mediate direct cytotoxicity against infected cells and contribute to the formation of syncytia. Additionally, natural killer (NK) cells and antibody production by B cells assist in eliminating the virus. Apoptosis of infected cells is facilitated through Fas/FasL interactions, highlighting the coordinated effort of the immune system to resolve the infection [68,69].

## 4. Potential Cancer Links of COVID-19

The correlation between SARS-CoV-2 infection and cancer risk is complex and under active investigation. COVID-19 can influence cancer formation and progression by both direct viral actions and indirect mechanisms such as immune system dysfunction and chronic inflammation. Chronic inflammation in long COVID, characterized by increased IL-6, TNF-α, and inflammasome activation, resembles pro-tumorigenic microenvironments; nevertheless, direct data connecting long COVID to de novo cancer start is scarce. Present hypotheses concentrate on extended immunological dysregulation (e.g., T-cell depletion) facilitating latent EBV reactivation. DNA damage induced by inflammation in chronically infected tissues.

In patients with long COVID, continuous inflammatory reactions may raise the chance of developing cancer over time, particularly through the accumulation of cellular damage and disrupted tissue homeostasis [70,71,72]. Dysregulation of the immune system brought on by SARS-CoV-2 infection can also significantly affect the body’s capacity to control viral replication and the growth of cancer cells. In one instance, the virus can aggravate autoimmune reactions, trigger cytokine storms, and impact T-cell activity. These immunological conditions may make it more difficult for the body to identify and get rid of cancer cells, producing an environment in which tumors may grow or advance more easily [73,74,75]. An overabundance of pro-inflammatory cytokines is a hallmark of cytokine storms, which may result in tissue damage and an environment that encourages tumor growth. In patients with pre-existing tumors, SARS-CoV-2 immune suppression can accelerate the course of these malignancies or, in certain cases, cause cancer relapse in previously stable patients. This immunological dysregulation may also reduce the efficiency of cancer therapies, as treatments like chemotherapy or immunotherapy may be delayed or weakened in the face of continuing viral infection [76,77,78,79,80]. Furthermore, the immunological suppression brought on by COVID-19 may encourage the reactivation of dormant viruses like EBV and human papillomavirus (HPV), which are connected to several cancer types [43,81,82,83].

These viruses may become more carcinogenic in infected people if they reactivate. Similar to this, changes in the SARS-CoV-2 virus itself, namely in immune-modulating proteins, may also change how the virus interacts with host immune systems, which could lead to the development of tumors. Although research into these pathways is ongoing, there is growing worry that the viral infection and the ensuing dysregulation of the immune system may cause new cancers to develop or hasten the course of already existing ones [83,84,85,86,87]. The complex relationship between the gut and the lung, known as the Lung-Gut Axis, emphasizes how viral infections like SARS-CoV-2 can cause immunological dysregulation and systemic inflammation (Figure 4). It specifically emphasizes how immunological dysregulation can cause latent infections like Human Herpesvirus 6 (HHV-6), Cytomegalovirus (CMV), and EBV to reactivate. The reactivation of these dormant viruses leads to microbial translocation, which also disrupts the balance of the gut microbiota, increases inflammation, and alters immune responses. This process damages endothelium and increases systemic inflammation, which worsens immunological dysregulation. The crucial roles played by important viral proteins, such as ORF8, S, N, and E proteins, which harm endothelial tissues and penetrate the blood-brain barrier to impact organ systems like the brain, heart, and kidneys, underscore the significant role that latent viral reactivation, including EBV, plays in amplifying systemic pathological effects through the Lung-Gut Axis [88,89,90,91].

SARS-CoV-2’s systemic disturbances and immune suppression may make it more difficult for the body to regulate tumor growth in people who have already received cancer treatment. SARS-CoV-2 can sometimes accelerate the development of malignancies that were stable or in remission before the infection. Additionally, failure to continue cancer therapies during acute infection or recuperation may result in adverse consequences, such as the advancement or recurrence of the disease [35,92,93,94,95].

## 5. Dual Threat of EBV and COVID-19 in Cancer Development: The Synergistic Effects

A well-known oncogenic virus, EBV is closely linked to the emergence of certain cancers, such as nasopharyngeal carcinoma, Hodgkin lymphoma, and Burkitt’s lymphoma. Concerns have been raised by recent research on the possible synergistic effects of EBV and COVID-19, especially in immunocompromised people like transplant recipients or patients with chronic COVID. Periods of immunosuppression have been shown to cause EBV reactivation, and COVID-19 infection may function as a catalyst for latent EBV reactivation, raising the risk of virally induced malignancies [14,22,96,97,98]. The immunological dysregulation brought on by COVID-19 may pose an extra concern to immunocompromised people, who are already more vulnerable to EBV-related cancers. Immune system changes brought on by COVID-19 infection, including cytokine storms and T-cell depletion, may compromise the immune surveillance systems required to regulate EBV reactivation and its carcinogenic potential [40,99,100,101]. In these patients, the interplay between EBV reactivation and COVID-19-related immune disruption could create an environment conducive to the development of EBV-associated cancers.

Moreover, significant alterations in the tumor microenvironment (TME) have been linked to COVID-19 infection, which may further promote the carcinogenesis linked to EBV. The COVID-19-induced inflammatory reactions may increase EBV replication, encourage viral latency, or contribute to a tumor-promoting environment (TME). This is especially important for tumors like nasopharyngeal carcinoma, which is intimately linked to EBV infection. In these cases, the virus can affect the surrounding tissue milieu as well as the host’s immune response. Tumor Promotion and Immune Evasion [37,43,102,103,104,105]. Both COVID-19 and EBV have evolved complex defense mechanisms against the host’s immune system, which may increase their capacity to induce carcinogenesis (Table 4). In order to evade identification and removal, the herpesvirus EBV can create latent infections inside immune cells, especially B lymphocytes, and alter immune responses [40,106,107,108,109,110]. Inducing immunological tolerance, which stops the host’s immune system from identifying and getting rid of EBV-infected cells, is one important way that EBV encourages the growth of tumors. The virus’s capacity to cause cancer is greatly influenced by this immune evasion [111,112,113,114,115].

COVID-19, on the other hand, interacts with the immune system in several ways, many of which may exacerbate EBV-associated malignancies. SARS-CoV-2, the causative agent of COVID-19, has been shown to impair the function of T-cells, B-cells, and natural killer (NK) cells—critical components of the immune system that play vital roles in immune surveillance against infections and malignancies. In particular, the COVID-19-induced immune suppression, characterized by lymphopenia and a dysfunctional immune response, may compromise the host’s ability to control latent EBV infections [31,98,121,122]. Moreover, a hyperinflammatory state that promotes the development and dissemination of EBV-infected cells may be a result of the cytokine storm frequently observed in severe COVID-19 patients. In addition to aiding in viral replication, elevated pro-inflammatory cytokines like TNF-α and IL-6 can also foster the growth of tumors. These immunological changes can lead to an increased risk of malignant transformation, especially in people already predisposed to EBV-related malignancies [40,123,124,125].

Additionally, there is evidence of a delayed or inadequate immune response in COVID-19 patients, which may make it more difficult to eradicate EBV-infected cells. This protracted immunosuppression may enable EBV to remain dormant and maybe encourage the malignant development of infected cells [43,106,126,127]. This dual immunosuppressive effect—both from the virus itself and from the COVID-19 infection—may accelerate the process of EBV-associated cancer progression (Figure 5) [103,128]. During acute COVID-19, an inflammatory response is initiated, which includes immune cell activation. This inflammatory milieu may promote the reactivation of latent EBV. Once reactivated, EBV can cause inflammation, resulting in a bidirectional interaction that maintains immunological activation. Persistent inflammation and immunological dysregulation can cause autoreactivity, in which autoantibodies attack the body’s own tissues. This cascade of events, triggered by the interaction of acute COVID-19, inflammation, and EBV reactivation, contributes to the pathogenesis of protracted COVID, emphasizing the complex immune-mediated mechanisms that underpin this illness [129,130].

## 6. Clinical Implications, Impact on Cancer Diagnosis and Treatment:

Global healthcare systems have been severely disrupted by the COVID-19 pandemic, which has affected the prompt diagnosis and treatment of a number of illnesses, including cancer. Regular screenings and timely diagnostic treatments are generally necessary for cancer detection, but many of these were postponed or delayed because of the pandemic’s constraints and the healthcare system’s preference for COVID-19 cases [35,131,132]. For patients with EBV-related cancers, these delays can be particularly harmful, as early detection is critical to improving survival outcomes. Moreover, the immunocompromised state induced by both EBV infections and COVID-19 creates additional complexities in the treatment of cancer. EBV has been shown to alter immune responses, promoting viral persistence and contributing to the malignancy of certain cancers. In addition, COVID-19 infection itself can weaken the immune system, further complicating the ability of cancer patients to respond to therapies effectively. These overlapping effects may result in poorer prognosis, delayed treatment responses, and overall worse outcomes for cancer patients during the pandemic [133,134,135,136]. The immune suppression brought on by COVID-19 may also increase the hazards for cancer patients with EBV-related cancers. The body’s capacity to fight cancer progression may be hampered by the immunological dysregulation seen in both disorders, which could lead to consequences and an increased disease burden [137,138]. Integrating oncology and virology research is critical to address EBV/COVID-19 synergies. Priorities include biomarkers for EBV reactivation and trials combining antivirals with immunotherapy.

## 7. Future Directions

Managing patients with EBV-related tumors during the continuing COVID-19 epidemic necessitates comprehensive, interdisciplinary treatments that address both viral infections simultaneously. Clinicians must be attentive in monitoring for possible EBV reactivation in COVID-19 patients, especially given the immunosuppressive nature of both diseases. Establishing treatment strategies that balance cancer medicines with adequate viral infection care is crucial for sustaining these patients’ health and well-being. Future studies should concentrate on a few critical areas to gain a better understanding of the intricate interactions between EBV, COVID-19, and cancer. One of the key research directions should be the study of viral reactivation mechanisms, particularly in the context of COVID-19-related immune regulation. Understanding how COVID-19 exacerbates EBV reactivation may aid in the development of targeted therapeutic options to prevent cancer progression in these vulnerable patients. Additionally, during the pandemic, studies on immune regulation and the discovery of biomarkers for outcome prediction in patients with EBV-related malignancies may enhance prognosis and direct more efficient treatments. Future studies on targeted medications that can simultaneously treat immunological alterations caused by viruses and cancer appear promising. Ultimately, the connection between EBV, COVID-19, and cancer emphasizes how urgently collaborative research projects are needed to develop improved techniques for the detection, control, and therapy of these dual-threat diseases.

Global problems brought on by the COVID-19 pandemic and EBV infections have significantly impacted the onset, diagnosis, and treatment of cancer. EBV, a well-known oncogenic virus, has long been implicated in the pathogenesis of a number of malignancies, and its relationship with COVID-19 further complicates the clinical management of cancer patients. The combined effects of COVID-19 and EBV can exacerbate tumor progression, impair immunological responses, and delay timely therapy since they share common immune dysregulation pathways. The COVID-19 pandemic has created additional challenges by delaying cancer diagnosis, disrupting treatment regimens, and creating new vulnerabilities for an already immunocompromised group of cancer patients with EBV. In addition to making it more difficult for the body to fight viral reactivation, the immune suppression caused by COVID-19 also lowers the efficacy of cancer treatments, which worsens the prognosis for these individuals. Treatment approaches need to adapt going ahead to address the combined threat of COVID-19 and EBV. Improving patient outcomes will need to incorporate multidisciplinary approaches to treat both viral infections at the same time while adhering to cancer treatment plans. In order to create targeted treatments that meet the unique immunological challenges presented by COVID-19, more research is required to comprehend the molecular processes allowing EBV reactivation in this setting. Last but not least, the connection between viral infection and cancer emphasizes the necessity of learning more about the interactions between viruses like EBV and contemporary global health issues like COVID-19. We can reduce the impact of these two dangers on cancer patients worldwide by developing more efficient diagnostic, treatment, and preventive methods through continued research and cooperation.

## 8. Conclusions

The COVID-19 pandemic and EBV infections are presenting concurrent global challenges that have significantly changed the development, diagnosis, and treatment of cancer. EBV, a well-known oncogenic virus, has long been implicated in the pathogenesis of a number of malignancies, and its relationship with COVID-19 further complicates the clinical management of cancer patients. COVID-19 and EBV share the same pathways of immune dysregulation, and when they work together, they can impair immune responses, hasten the growth of tumors, delay timely treatment responses, and postpone prompt medical attention.

The COVID-19 pandemic has created additional challenges by delaying cancer diagnosis, disrupting treatment regimens, and creating new vulnerabilities for an already immunocompromised group of cancer patients with EBV. In addition to making it more difficult for the body to fight viral reactivation, the immune suppression caused by COVID-19 also lowers the efficacy of cancer treatments, which worsens the prognosis for these individuals.

Future treatment approaches must be modified to address the combined threat of COVID-19 and EBV. Improving patient outcomes through the use of interdisciplinary techniques to treat both viral infections at the same time while safeguarding cancer treatment regimens will be crucial. More research is needed to understand the molecular pathways underlying EBV reactivation in the context of COVID-19 and to develop tailored therapies that consider the unique immunological challenges presented by both illnesses. Ultimately, this link between viral infection and cancer emphasizes the need for a deeper understanding of how viruses such as EBV interact with contemporary global health crises like COVID-19. Through continued study and collaboration, we can develop more effective diagnostic, therapeutic, and preventive methods to reduce the burden of these concurrent risks on cancer patients worldwide.

## Figures and Tables

**Figure 1 viruses-17-01195-f001:**
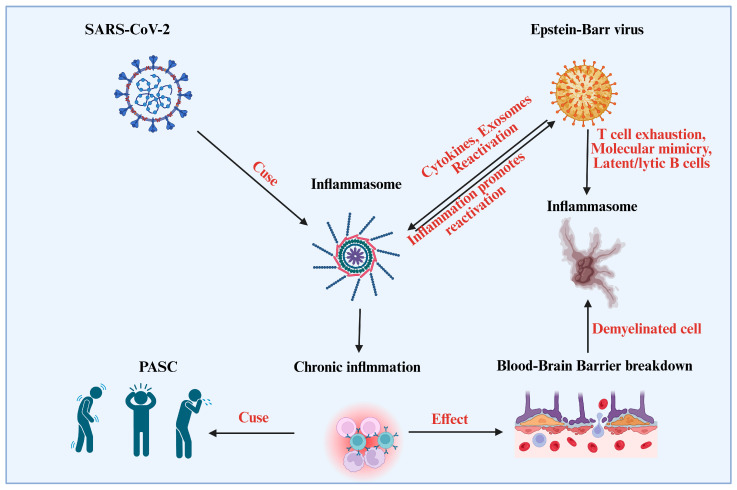
Mechanisms Linking SARS-CoV-2, EBV Reactivation, and Post-Acute Sequelae of COVID-19 (PASC). The figure showed how immune responses like T cell exhaustion and molecular mimicry interact with viral infections like Epstein-Barr virus and SARS-CoV-2. It draws attention to effects on B cells, chronic inflammation, and inflammasome activation. Among the repercussions are demyelination and disruption of the blood-brain barrier, which may be a factor in Post-Acute Sequelae of SARS-CoV-2 (PASC). The intricate connection between immune systems, viral infections, and long-term health consequences is highlighted in the diagram.

**Figure 2 viruses-17-01195-f002:**
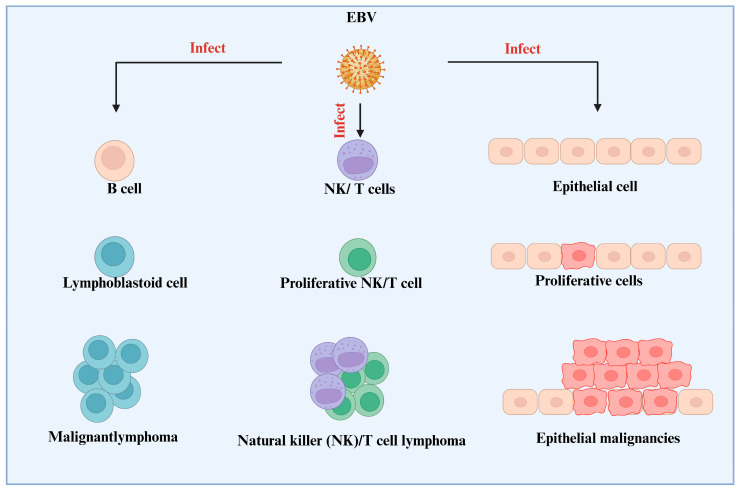
EBV-Associated Malignancies: Cell-Specific Transformation and Lymphoproliferative Disorders. This figure explains how the EBV interacts with different cell types, such as B cells, NK/T cells, and epithelial cells. It draws attention to how proliferative NK/T cells and lymphoblastoid cells can change and develop into cancer. The graphic highlights the emergence of cancers such as epithelial cancers and natural killer (NK)/T cell lymphoma. It emphasizes how EBV promotes cancer and cellular proliferation. This illustration links the pathophysiology of several malignancies to EBV infection.

**Figure 3 viruses-17-01195-f003:**
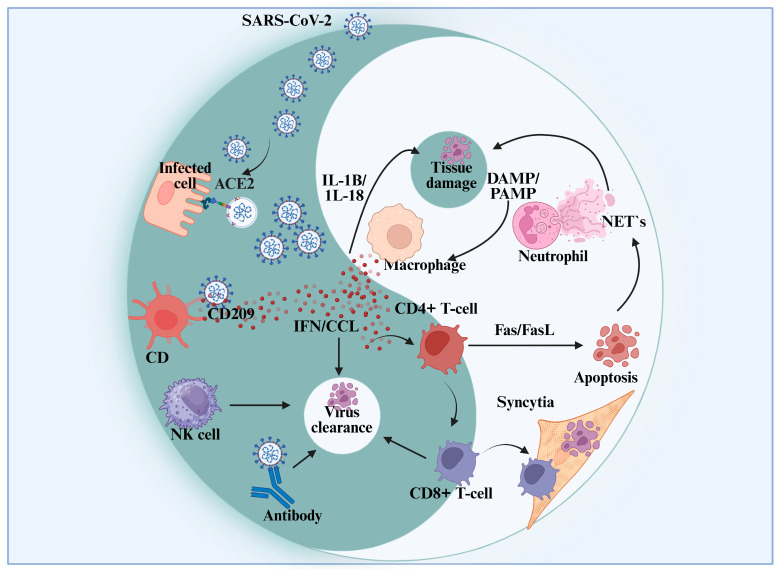
SARS-CoV-2 Immune Response: Viral Entry, Immune Activation, and Tissue Injury. The figure depicts the immunological response to SARS-CoV-2 infection, emphasizing important elements that cause inflammation, such as PAMPs and DAMPs. It demonstrates how ACE2 receptors, which involve neutrophils, NETs, and macrophages, contribute to viral entry and subsequent tissue damage. The figure shows how NK cells, CD4+ and CD8+ T-cells, and antibody production are activated to eliminate viruses. Additionally, it shows how infected cells undergo apoptosis and develop syncytia. The intricate relationship between viral infection and host immunological responses is highlighted in this figure.

**Figure 4 viruses-17-01195-f004:**
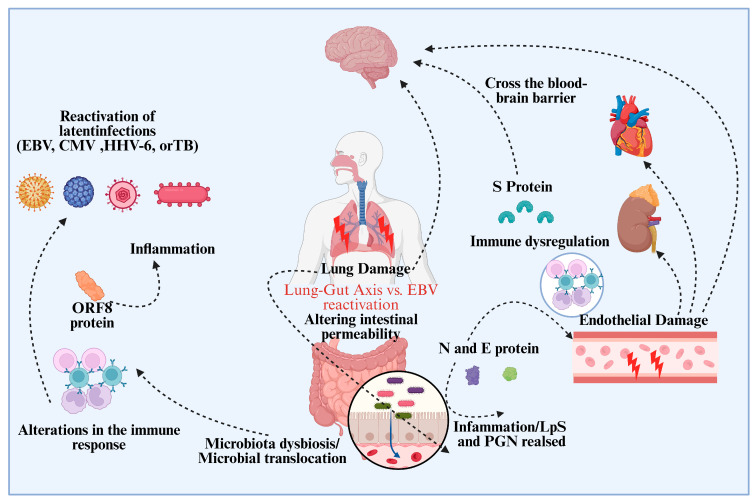
Mechanisms of COVID-19 Pathogenesis: Viral Reactivation, Systemic Inflammation, and Multi-Organ Damage. This figure demonstrates SARS-CoV-2’s systemic effects, with a focus on its capacity to pass the blood-brain barrier and reawaken latent diseases such as EBV, CMV, HHV-6, and tuberculosis. It emphasizes immunological dysfunction and inflammation produced by viral proteins such as S, N, and E proteins. The graphic depicts lung damage and the lung-gut axis, demonstrating how the virus affects intestinal protein permeability and induces endothelium damage. It also focuses on microbiota dysbiosis, microbial translocation, and the release of inflammatory chemicals such as LPS and PGN. This image emphasizes SARS-CoV-2’s extensive influence on many organ systems and immunological responses.

**Figure 5 viruses-17-01195-f005:**
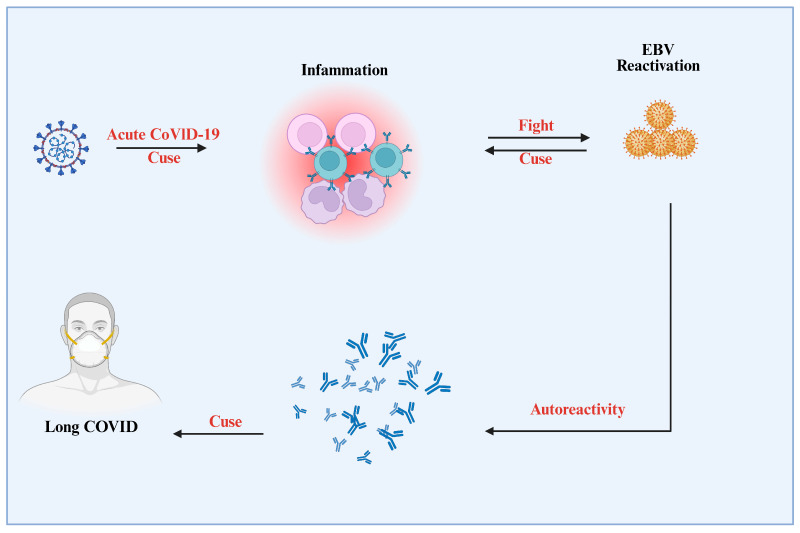
Potential Pathways Linking Acute COVID-19, EBV Reactivation, Inflammation, Autoreactivity, and Long COVID. This figure illustrates hypothesized mechanisms connecting acute COVID-19 infection to Long COVID, including Epstein-Barr virus (EBV) reactivation, systemic inflammation, and autoimmune responses (autoreactivity). The arrows suggest possible interactions or progression between these factors.

**Table 1 viruses-17-01195-t001:** The comparison between EBV and COVID-19 (caused by SARS-CoV-2) based on their characteristics, effects on cancer, and related factors.

Aspect	EBV	SARS-CoV-2
Type of Virus	Herpesvirus (HHV-4), oncogenic virus [20]	Coronavirus (SARS-CoV-2), recent viral infection [21]
Prevalence	Affects 90–95% of people worldwide [22]	Newer virus with widespread global impact [23]
Transmission	Primarily through saliva (“kissing disease”) [24]	Primarily through respiratory droplets, close contact [25]
Common Infections	Frequently asymptomatic, can cause mononucleosis (“mono”) [26]	Can range from asymptomatic to severe respiratory illness [27]
Cancer Associations	Strongly associated with certain stomach malignancies, Burkitt lymphoma, Hodgkin lymphoma, and nasopharyngeal carcinoma [4]	Not directly associated with cancer, but research into long-term cancer risks is ongoing [28]
Cancer Development Mechanism	Infects B-cells, promotes immune evasion, genetic instability, and cancer [29]	Immunosuppression and inflammation may increase cancer risk over time, particularly in high-risk individuals [19]
EBV Reactivation Risk	Reactivation in B-cells under specific conditions (immunosuppressive states) can lead to cancer [30]	May interact with latent infections like EBV, potentially leading to reactivation and cancer formation [31]
Immunological Impact	Latency in B-cells allows immune evasion [32]	Causes significant immune dysregulation, especially in severe cases or during treatment [33]
Impact on Cancer Care	Known viral oncogenic effects with established diagnostic protocols [34]	Delays in cancer care during the pandemic could affect cancer diagnosis and treatment, potentially worsening outcomes [35]
Chronic Inflammation	Can contribute to chronic inflammation in cancer types associated with EBV [36].	Chronic inflammation in long COVID could contribute to increased cancer risk [37].

**Table 2 viruses-17-01195-t002:** EBV Oncogenic Mechanisms and Therapeutic Targets.

Mechanism	Key Proteins	Pathways Affected	Associated Cancers	Therapeutic Approaches
B-cell immortalization	LMP1, EBNA2	NF-κB, JAK/STAT	Burkitt lymphoma	CD19 CAR-T therapy
Immune evasion	EBNA1, BPLF1	Antigen presentation	Hodgkin lymphoma	PD-1 inhibitors
Viral latency	EBERs, LMP2A	B-cell signaling	Nasopharyngeal carcinoma	EBV-specific CTLs
Inflammation	LMP1	Cytokine production	Gastric cancer	Anti-IL-6 therapies

**Table 3 viruses-17-01195-t003:** Mechanisms of SARS-CoV-2 Infection and Its Impact on Host Functions.

Mechanism	Details
Virus Type and Characteristics	SARS-CoV-2 is an RNA virus with a spike protein that facilitates entry into cells via ACE2 receptors, primarily targeting respiratory epithelial cells [58,59,60].
Viral Entry and Replication	The spike protein interacts with ACE2, enabling endocytosis and replication inside the host cell [60,61].
Impact on Host Functions	SARS-CoV-2 affects immune responses, inflammatory signaling, and autophagy, contributing to disease severity and potentially long COVID [62,63,64].
Target Organs	Primarily the lungs, but it can also affect the heart, kidneys, and intestines [65,66].

**Table 4 viruses-17-01195-t004:** Impact of COVID-19 on EBV Reactivation and Tumor Development.

Aspect	Details
EBV Reactivation Risk	Immunosuppression caused by COVID-19 may catalyze latent EBV reactivation, raising the risk of virally induced malignancies in immunocompromised patients [72,116,117].
Impact on Immune Surveillance	COVID-19-induced dysregulation, including cytokine storms and T-cell depletion, compromises immune surveillance, enabling EBV to reactivate and promote carcinogenesis [118].
Tumor Microenvironment Changes	COVID-19-induced inflammation may promote EBV replication and latency, creating a tumor-promoting microenvironment, especially for NPC [31,119].
Immune Evasion by EBV	EBV can evade immune detection by inducing immune tolerance and latent infections in immune cells, which supports tumor growth [72,116].
Impact on COVID-19 Immune Responses	COVID-19 impairs T-cells, B-cells, and NK-cells, reducing the ability to control latent EBV infections and promoting tumor development [76,120].

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
