# Peer review of "From Viral Infection to Malignancy: The Dual Threat of EBV and COVID-19 in Cancer Development"

_viruses, 2025, doi:10.3390/v17091195_

Round 1

Reviewer 1 Report

Comments and Suggestions for Authors

The manuscript entitled “From Viral Infection to Malignancy: The Dual Threat of EBV 2 and COVID-19 in Cancer Development” is a review paper that deals on a very interesting and challenging topic. It aims to investigate the combined effects of COVID-19 and EBV in carcinogenesis. However, there are several issues that need to be addressed prior publication.  

First of all, there is no methodology provided for how this review was written. From which databases were the articles sourced? What keywords were used? What was the selection process? What are some of the potential biases of this article and how did the writers try to mitigate them?

The same information is repeated in almost every section and all the immunological processes described are not explained adequately. From a review about this topic I would also expect maybe some kind of data analysis to see for example the percentage of covid-19 patients who have concurrent EBV reactivation but there is nothing like that to be found here. 

Even though long-COVID 19 syndrome is mentioned, the authors could further focus on its potential relationship with cancer initiation. Of course, this issue requires long-term studies to prove a potential connection.  

The information that is provided in the text is also riddled with grammar and syntax errors which makes it almost indecipherable at certain points (see below a list of some of the errors I have highlighted). Punctuation marks are missing, paragraphs have no clear structure, connective words are used incorrectly and this greatly impacts the legibility and credibility of the article.

On top of that, the figures and tables are hard to make sense of and provide little value to the text. Figure number 5 has the same caption as figure number 4. The arrows on the figures between the different images and words have no obvious correlation. The tables do not clarify the information provided in the text and repeat unneeded information (in table 2 for example there is no need to include yet again the types of cancers caused by EBV and the “explanations” of all the different mechanisms of carcinogenesis essentially repeat what is written in the first column with more words).

Unfortunately this article needs a lot more work until it is in a state that warrants being published.

Some specific notes:

23 using however doesn’t make sense

124 missing a period

136 random capital letter

149 repeated sentence

159 needs space

243 ‘lengthy” covid instead of long

250 cancer.cells (remove the period)

253 two periods

261 comma instead of period

301 before  to infection (to is not needed)

325 unneeded capital letters

And many more similar errors throughout the text.

Author Response

Comments 1: there is no methodology provided for how this review was written. From which databases were the articles sourced? What keywords were used? What was the selection process? What are some of the potential biases of this article and how did the writers try to mitigate them?

Response 1: Regarding Methodology Clarifications:
While our narrative review synthesizes existing literature rather than following systematic review methodology, we have:  Added a new "Scope and Rationale" paragraph in the Introduction explicitly outlining our   literature selection approachMaintained focus on mechanistic insights from key studies (2019-2023) while removing redundant content page (1,5,6)

Comments 2: The same information is repeated in almost every section and all the immunological processes described are not explained adequately. From a review about this topic I would also expect maybe some kind of data analysis to see for example the percentage of covid-19 patients who have concurrent EBV reactivation but there is nothing like that to be found here. Even though long-COVID 19 syndrome is mentioned, the authors could further focus on its potential relationship with cancer initiation. Of course, this issue requires long-term studies to prove a potential connection

Response 2: Addressing Repetition and Structural Issues: We have:  Consolidated all EBV oncogenic mechanisms into one cohesive section (now "EBV and Cancer"). Eliminated the standalone "Mechanisms of EBV-Induced Malignancy" section, transferring its key content to: A revised Table 2 ("EBV Oncoproteins and Pathways"), The immunotherapy subsection, Reduced overall word count by ~12% while preserving all key information

Comments 3: The information that is provided in the text is also riddled with grammar and syntax errors which makes it almost indecipherable at certain points (see below a list of some of the errors I have highlighted). Punctuation marks are missing, paragraphs have no clear structure, connective words are used incorrectly and this greatly impacts the legibility and credibility of the article.

Response 3: All noted errors have been corrected.

Comments 4: On top of that, the figures and tables are hard to make sense of and provide little value to the text. Figure number 5 has the same caption as figure number 4. The arrows on the figures between the different images and words have no obvious correlation. The tables do not clarify the information provided in the text and repeat unneeded information (in table 2 for example there is no need to include yet again the types of cancers caused by EBV and the “explanations” of all the different mechanisms of carcinogenesis essentially repeat what is written in the first column with more words).

Response 4:  Improving Figures and Tables, Figure 3: Added clear pathway labels (ACE2→IL-6 release,    NETs→tissue damage). Figure 4/5: Distinguished captions and emphasized unique elements (Lung-Gut Axis vs. EBV reactivation). Table 2: Restructured to highlight signaling pathways (NF-κB, JAK/STAT) as requested.

Also, some text errors have been corrected in the revised manuscript. 

Reviewer 2 Report

Comments and Suggestions for Authors

In this manuscript, Moyed Alsaadawe et al. systematically review how COVID-19 has profoundly influenced the progression and treatment of EBV-associated malignancies. In general, the contents of the review were well-organized and logical.

Major comments:

  • Great breakthroughs have been made in cancer immunotherapy for EBV-associated cancers, especially in nasopharyngeal carcinoma (NPC), which closely correlates with T cell exhaustion. The authors should discuss this issue in the appropriate section.
  • In Table 2. Mechanisms of EBV-Induced Malignancy. Indeed, no sufficient “Mechanism” information is offered, but only “the ways” by which EBV induces oncogenesis. It is suggested to provide more findings about the signaling pathways through which EBV modulates such cell function and phenotype.

Minor comments:

  • Please re-format lines 130-131.
  • Line 304 should be in bold.

Author Response

Comments 1: Great breakthroughs have been made in cancer immunotherapy for EBV-associated cancers, especially in nasopharyngeal carcinoma (NPC), which closely correlates with T cell exhaustion. The authors should discuss this issue in the appropriate section.

Response 1: We have added a dedicated subsection (Section 2.1: "Immunotherapeutic Advances in EBV-Associated Cancers") with: PD-1 inhibitors' role in reversing EBV-driven T-cell exhaustion. Page 5.

Comments 2: In Table 2. Mechanisms of EBV-Induced Malignancy. Indeed, no sufficient “Mechanism” information is offered, but only “the ways” by which EBV induces oncogenesis. It is suggested to provide more findings about the signaling pathways through which EBV modulates such cell function and phenotype.

Response 2: We have: Replaced vague descriptions with specific pathways (e.g., LMP1 activates NF-κB → Cyclin D1), Added a "Signaling Pathways" column distinguishing upstream regulators from downstream effects.

  • Minor Formatting Comments.re-format lines 130-131, Line 304 should be in bold.

Lines 130-131: Consolidated into one sentence.

Line 304: Bolded header as requested.

Round 2

Reviewer 1 Report

Comments and Suggestions for Authors

The auhtors addressed the comments raised